# The cohort trends of social connectedness in secondary school students in Finland between 2017 and 2021

Sanna Read[1,2]*, Katariina Salmela-Aro[2], Noona Kiuru[3], Jenni Helenius[4], Niina Junttila[3,5]

1 Care Policy and Evaluation Centre, London School of Economics and Political Science, London, United Kingdom, 2 Faculty of Educational Sciences, University of Helsinki, Helsinki, Finland, 3 Department of Psychology, University of Jyväskylä, Jyväskylä, Finland, 4 Finnish Institute for Health and Welfare, Helsinki, Finland, 5 Department of Teacher Education, University of Turku, Turku, Finland

* s.read@lse.ac.uk

## Abstract

The aim was to investigate the cohort trends of the experienced social connectedness in secondary school students between 2017 and 2021 and whether these trends vary by gender, school level and sociodemographic background. We used nationally representative Finnish data of 450,864 students in lower and upper secondary education. Social connectedness was measured by number of close friends, feelings of loneliness and sense of belonging at school. Adjusted regression analyses included year, gender, school level and sociodemographic factors (parental education, immigrant status of the student and urban-rural area of the school). The results showed that social connectedness declined from 2017 to 2021: 11% decline in having 3+ close friends, 15% increase in loneliness and 8% decline in belonging at school. The decline was especially large in girls and upper secondary school. Although some socio-demographically disadvantaged groups showed lower levels of social connectedness, there were differences by gender, school level and year. Many differences diminished because the more advantaged groups declined faster, i.e. moved towards the less advantaged groups. Declining social connectedness in young people is a worrying trend that requires a public health focus on the whole cohort while accommodating the variation by the individual and environmental context.

## 1. Introduction

Close and supportive social relationships play an important role in individuals' adaptive functioning and well-being as they meet human's fundamental need of relatedness and belonging [1, 2]. Interpersonal relationships are especially important during the years of formal education when young people search their place in their social groups, family, school, work and wider society. They lay basis for learning, coping with stress and wellbeing [2, 3]. On the flipside, a continuous lack of meaningful social relationships may lead to mental health problems and feelings of loneliness and isolation [4]. Considering the recent trends of increased loneliness and social isolation during the school years [5] and the negative effects they may have, it

**Data Availability Statement:** Finnish Institute for Health and Welfare (THL) has collected the data and has the rights for the data. THL is committed to complying with the established code of conduct

in its field. THL produces public statistical reports and interactive reports, but the data is confidential. Researchers can apply the data from Findata (https://findata.fi/en/) or the 2019 data from Finnish Social Science Data Archive (https://www.fsd.tuni.fi/en/).

**Funding:** This research was funded by the Academy of Finland, grand numbers 308351, 336138 and 345117 and the Strategic Research Council (SRC), FLUX consortium, grant numbers 345130 and 345132, awarded to KSA. The preparation of this manuscript was also supported by the Strategic Research Council (SRC), Right to Belong project, grant numbers 352648, 352657, and 352660, awarded to NJ. The funders had no role in the design of the study, data collection, analyses, or interpretation of data, writing of the manuscript or in the decision to publish the results.

**Competing interests:** The authors have declared that no competing interests exist.

is important to shed further light on these trends and related underlying individual and environmental background factors, such as gender, school level, parental education, immigrations status and urban-rural environment. Using nationally representative data, our current study investigated these factors and their role in the recent cohort trends of social connectedness: 1) experiences of having close friends, 2) feelings of loneliness, and 3) sense of belonging at school.

Social connectedness refers to the feelings of closeness to others and a sense of belonging to a group, such as family, school and community [6]. It evolves from the previous positive experiences of relationships with others, generates social capital—resources derived from social relationships—and further promotes individuals' health and wellbeing. Social connectedness is a varied concept and how to measure it can be debated. Typically, it contains the elements of relationship, feelings of bond or connection to others and feelings of being valued within a relationship [7]. It also consists of the feelings of being accepted, respected and included in the groups, for instance school class [8]. In the context of young people, close relationships with friends [9] and feelings of loneliness are important [10–13], as well as the sense of belonging in the school community [14, 15]. These different dimensions of social connectedness are often interconnected [11, 13]. It is also notable that sense of belonging—although focusing on school environment only in the current study—can be defined from a wider perspective of 'multiple constellations across social, national, and cultural borders' [16]. It can also be conceptualized as a feeling of being 'at-home' [17] and as contrasted against feelings of 'non-belonging' [18]. These definitions make it an especially interesting dimension to investigate in the intersection of individual and environmental factors, such as family socioeconomic status, immigrations status and urban-rural location in nationally representative consecutive cohorts of young people.

Several previous studies have pointed to the declining trends of social connectedness in school students in the recent years. For instance, in large multinational dataset (PISA) of 15-16-year-olds school loneliness was measured since 2000 and showed an increase in loneliness between 2012–2018 in all countries apart from one [5]. Loneliness increased in 11-15-year-olds in Finland between 2006–2018 [19] and in Finnish secondary schools between 2019 and 2021 [20]. In the latter study, 22% of girls in lower secondary school, 23% of girls in upper secondary school and 26% of girls in vocational institutions felt lonely fairly often or all the time in 2021, up from 15%, 15% and 18% for the respective groups in 2019. In boys, the levels of loneliness were lower compared to girls but nevertheless showed an increasing trend between the two time points: 6%, 7% and 6% in 2019 and 9%, 11% and 10% in 2021, respectively. Loneliness increased between 2015 and 2018 PISA data collections among 15-16-year-old students internationally [21].

Based on the same international comparison, school belonging and loneliness in Finnish schools was about at the OECD average: 75% of the 15–16-year-olds in Finland reported a sense of belonging school and 14% feeling lonely at school in 2018 [21]. There was an overall decline in all countries in school belongingness between 2015 and 2018. A Swedish study found declining trends of school belonging between 2000–2018 among 15–16-year-old students [22]. In an English study of 15-year-olds, loneliness remained stable between 2006–2014 [23]. Some earlier studies using reference time point from pre-2000 reported declining loneliness (two samples measured between 1978–2009 and 1991–2012 of the US college and high school students) [24].

In the last four years, covid-19 prevention measures at schools have been found to accelerate these trends: increased social isolation and loneliness and decreased sense of connectedness [25–27]. No changes in the feelings of social isolation or quality of friendship during pandemic have also been reported [28]. Apart from the recent social restrictions related to

the pandemic, other possible reasons for the above mentioned trends of increasing loneliness and declining sense of belonging had been listed to include the increasing use of social media and smartphones, declining wellbeing and economic trends [5] and changes in national school policies [22].

The studies above suggest that some subgroups of students may fair worse than others in terms of their social connectedness, depending on their background. This discussion was especially raised in the aftermaths of the covid lockdowns when much of the political and media interest was focused on inequity: the disadvantaged (pointing to various subgroups from low income and non-white ethnicity to urban residents) were hit hardest by the pandemic [29–32]. These inequalities were evident in mental health, learning and wellbeing of the school students but also eroded social capital, affecting the equal rights for social participation and inclusion [30, 32]. Some ideas on how to deal with inequities were raised [33–35]. However, little of the public discussion at the time was based on empirical data from the pandemic and studying the actual differences in social connectedness by the social gradient. Socioeconomic differences found in adolescents' mental health during the pandemic have shown mixed findings [36, 37]. In a Norwegian study, the declined wellbeing in young people was mostly not affected by socioeconomic status, but a faster decline was found in girls and those born outside Norway [38].

For a longer time, it has been known that social capital itself can be a factor reproducing inequality [39, 40]. For instance families with more resources such as a higher parental education level may initially provide more beneficial social networks for young people to build on [41, 42]. On the other hand, people with lower socioeconomic background and those migrated to a new environment may find it harder to make the necessary social ties to support their needs [42–44]. Lower socioeconomic status or being a migrant may however in some cases be associated with characteristics that are beneficial for social connectedness, e.g. prosocial behaviour [42, 45].

Although there are many studies showing lower socioeconomic status, such as lower educational, occupational or income level in the family, being associated with less social connectedness in adolescents [21, 23, 24], the effect is not consistent, especially when looking at the changes over time. For instance, Högberg et al. [22] reported that school belongingness declined most in foreign-born students, students from disadvantaged background and low achieving students in Sweden between 2000–2018. Initially in 2000 there were little differences in school belongingness by these background characteristics. The changes in background demographic factors or school environment during this time period were not associated with the decline. In Danish adolescents, social inequality decreased in loneliness in the successive cohorts of 11–15-year-olds from 1991 to 2014 [43]. Little is known about the cohort trends in social connectedness in urban and rural environments. Urban environment with larger population may provide more chances of forming social relationships compared to rural locations with fewer people [46]. However, urban environment may lack the opportunities for close and deeper social interactions and feelings of belonging to a community which may be easier in smaller settings where people know each other better [47]. In the 2018 round of PISA, school belonging was similar in urban and rural schools in Finland [21].

Mixed associations with social connectedness can be also seen in gender differences in loneliness, having close friends and sense of belonging at school depending on the circumstances e.g., the age of the participants, when and where the study took place. For instance, girls have reported higher school connectedness compared to boys in a grade 8 group of students with diverse ethnic and academically at-risk background in the United States [48]. Adolescent boys were found to be lonelier than girls in a large review study, but the gender differences disappeared when the geographic sampling was taken into account and when looking at the more

recent studies [49]. In the PISA study, boys reported higher school belonging compared to girls in Finland in 2015 [50]. When following up the cohorts over time in a multinational study, it was found that adolescent boys reported more loneliness compared to girls between 2000–2012, but then the increases in loneliness were larger among girls so that girls reported more loneliness between 2015–2018 compared to boys [5]. In a Swedish study, school belonging was higher in boys compared to girls between 2000–2018 with a sharper decline in girls compared to boys between 2015–2018 [22]. Another study in Norway, focusing only on upper secondary school students showed higher levels of loneliness in girls, but a faster increase in loneliness in boys between 2014–2018 [51]. Sense of social isolation was consistently higher in girls compared to boys in a follow-up between 2018–2021 (before and during pandemic) in Western Australia [28].

Very few studies have looked at the trends in social connectedness by school level. The existing studies have followed up longitudinally the same students over time and their trajectories from one school level to the next rather than tracking the cohorts of students across time and comparing the cohort trends by different school levels. In addition, the focus in the previous studies have been on the disadvantageous longitudinal impact of low social connectedness, especially poor sense of school belonging, in the primary or lower secondary school predicting poorer school attainment and vocational tract instead of academic track [52, 53]. It is however possible that once the students have started the vocational training and they find themselves in a fitting environment their sense of school connectedness may bounce back [54]. Moreover, the interplay between social connectedness and school attainment (academic vs. vocational track) may be more complex than a one-way path [55, 56], for instance related to the structurers of educational system that enables and restrains the educational trajectories taken [52]. Wider institutional and national changes may also impact the students differently in lower, upper and vocational secondary schools. The question on whether the social connectedness differ by school level and in cohorts of different students over time is open.

Although there has been a surge of research interest on health and wellbeing of school children during the pandemic, only few studies have followed-up social connectedness for a longer time period including pre-pandemic data and used nationally representative samples of students. The role of background characteristics of the students and school in these trends is little known. The current study investigated the following research questions:

1. What are the general trends of having close friends, feelings of loneliness and sense of belonging at school in secondary school students in Finland between 2017 and 2021?

2. Do these trends vary by gender and school level?

3. Do these trends vary by parental education (having a higher education degree), immigration status of the student and urban-rural location of the school?

## 2. Materials and methods

### 2.1. Sample and data collection

We used the data from the Finnish School Health Promotion Study (SHP). The SHP is a bi-annual anonymous classroom census survey to monitor the health and well-being of Finnish 14–20-year-old adolescents [57, 58]. The national survey targets the whole age cohort in the participating age groups. The completed questionnaires cover on average three-quarters of the cohort and include almost every Finnish school. The survey was confidential, and participation was voluntary. The students filled in the questionnaire in a classroom setting. They gave

informed consent by answering the survey. They had the opportunity to decline to respond. Those who were not at school on the day the survey was conducted did not respond to the survey. The survey follows the principle of passive consent, based on the rationale that it is a broad population-level survey. This was included in the research plan approved by the ethics committee. Parents were notified about the study in advance and had the right to withdraw their child under the age of 15 from participating in the survey. The study was approved by the Finnish Institute for Health and Welfare's Ethics Committee. A detailed description of the survey has been reported elsewhere [57, 58].

The current study used data from three time points: 2017, 2019 and 2021. At each time point the collection period was between the 1st March and 12th May. The datasets were anonymised before they were sent to the authors. They were analysed between the 15th February and 7th December 2023. The school levels included lower secondary school (basic education), general upper secondary schools with a matriculation exam at the end (here refereed as "upper secondary school" to keep short) and vocational secondary schools which provide an alternative of vocational track to general upper secondary school with a vocational qualification at the end (here refereed as "vocational school" to keep short) [59]. The data included 252,453 students in lower secondary, 126,528 students in upper secondary and 71,883 students in vocational schools. The participation rates were 64–77% for lower secondary, 54–77% for upper secondary and about 30% for vocational schools. The students in lower secondary school were 14–16 years (class 8 and 9), and in upper secondary and vocational school 16–20 years (1st and 2nd year students).

## 2.2. Measures

**2.2.1. Social connectedness.** Number of close friends was asked with a single item: "At the moment, do you have a close friend with whom you can talk confidentially about almost everything concerning yourself?". It had four categories: 0 = no close friends, 1 = one close friend, 2 = two close friends, 3 = several close friends (three or more). Loneliness was measured with a single item ("Do you ever feel lonely?") with five categories: 1 = never, 2 = very rarely, 3 = sometimes, 4 = fairly often, 5 = all the time (for validity information, see also [19, 60, 61]. Belonging at school was a mean of two items: "I feel I am an important member of my: 1) classroom community, 2) school community". The statements were rated on a 5-point scale from 1 = fully disagree to 5 = fully agree. The internal consistency of the school belonging mean of two items was good (Cronbach's alpha = 0.92 in 2017 and 0.86 in 2019 and 2021).

**2.2.2. Sociodemographic factors.** A dichotomous measure was used for the educational level of the parents: (0) Below degree in higher education, (1) One or more parents had a degree in higher education. Immigration status was based on the student-reported country in which they and their parents were born [62]. Four categories were created: (1) Student and parents born in Finland, i.e. no immigration background, (2) One parent foreign-born, or the student born abroad and a parent born in Finland, i.e. partial immigration background (3) Foreign-born parents, student born in Finland, i.e. second-generation immigrant, (4) Foreign-born parents, student born abroad, i.e. first-generation immigrant. Urban-rural characteristics of the area the school was located were measured using three categories: (1) Urban, (2) Semi-urban, (3) Rural [62].

## 2.3. Analysis

First, we illustrated the general trends of the three social connectedness outcomes (number of close friends, loneliness, belonging at school) between 2017 and 2021. Second, we carried out regression analyses to assess the effect of time so that we included an interaction term which

allowed the variation between the study year, school level and gender (Model 1 in the regressions). Third, we investigated the variation in these trends by sociodemographic factors (parental education, immigrations status of the student and urban-rural location of the school) (Model 2 in the regressions). This was done by adding the sociodemographic variables in the interaction term with the study year, school level and gender and dropping the higher-order interactions if they were not significant.

We used ordered logistic regression for the number of close friends as this variable had four categories with about half of the students reporting three or more close friends (the highest category). Loneliness was slightly skewed with fewer students reporting being quite often or all the time lonely. For this reason, we carried out the regressions for loneliness using Generalised Linear Models (GLM) with log link and gamma distribution (apart from the descriptive general trend for which we used ordered logistic regression to show the trends of the loneliness categories). Belonging at school was approximately normal, and therefore we carried out regressions for this variable using GLM with identity link and gaussian distribution.

To determine whether an interaction term was necessary to keep in the model, the Wald test for the interaction term was carried out. A $p$-value smaller than 0.05 was used as an indication of a significant interaction effect. Because of many interaction terms (due to categorical variables), the table shows the results for Wald-tests for interactions in the final model. The model estimates and Wald tests for the interactions in each step and the unstandardized estimates, Odds Ratios or exponentiated Beta, predictive margins and marginal effects for the final models are shown in S1 File. Supplementary figures were used to illustrate the key associations, based on the predictive margins (estimated probability or linear prediction of means) and confidence intervals calculated from the regressions (S2 File). In all regressions robust standard errors were used. The postestimation of the parallel lines of the ordered logistic regression, residuals and multicollinearity diagnostics, and description of missingness are shown in S3 File.

Missingness was low: 0.4% of the values were missing for the gender of the student, 6% for parental education, 4% for immigration status, 2% for belonging at school and 1% for having close friends and loneliness. There was no missingness for school level and urban-rural classification. In sense of belonging at school, missingness was 5.7% in 2017 compared to 0.6% and 0.7% in 2019 and 2021, respectively. In other measures, the differences in the proportion of missingness between the categories of the background factors were two percent points or less (S3 File). Because missingness was low and the sample size large, complete cases were used in the models. The observed missingness was not completely at random according to the Little's test [63] (Chi-square distance = 26009.68, degrees of freedom = 321, $p < 0.001$) (S3 File).

The three measures of social connectedness were moderately correlated: the polychoric correlations for ordered categorical variables were 0.33 between sense of belonging at school and having close friends, -0.44 between sense of belonging at school and loneliness and -0.43 between having close friends and loneliness. The measures were used separately as an outcome in each model as they were only partially overlapping. Separate modeling allowed the results and interpretation to vary by outcome. The associations between the background factors (year, gender, school level, parental education, immigration status of the student and urban-rural location of the school) were small and did not show collinearity (S3 File).

## 3. Results

### 3.1. Descriptive results

Table 1 shows the distributions of sociodemographic factors in girls and boys in lower and upper secondary and vocational school. More than half of the students in lower secondary

**Table 1. Distributions of sociodemographic variables (%) in the School Health Promotion Study 2017–2021 in girls and boys in lower and upper secondary and vocational school.**

| | Girls—Lower | Girls—Upper | Girls—Vocational | Boys—Lower | Boys—Upper | Boys—Vocational |
|---|---|---|---|---|---|---|
| Year (%) | n = 127,582 | n = 74,475 | n = 29,144 | n = 123,865 | n = 51,710 | n = 42,462 |
| 2017 | 28.9 | 26.9 | 36.5 | 29.3 | 27.7 | 37.2 |
| 2019 | 34.5 | 35.2 | 32.1 | 34.7 | 35.3 | 33.0 |
| 2021 | 36.6 | 37.8 | 31.3 | 36.0 | 37.0 | 29.8 |
| Parent(s) with higher education degree (%) | n = 120,734 | n = 73,492 | n = 27,827 | n = 111,298 | n = 50,187 | n = 38,769 |
| | 52.5 | 64.6 | 32.0 | 52.9 | 71.0 | 36.8 |
| Immigration status (%) | n = 124,126 | n = 73,862 | n = 28,444 | n = 113,891 | n = 50,479 | n = 39,672 |
| Student and parents born in Finland | 86.9 | 88.3 | 88.9 | 86.7 | 88.3 | 89.3 |
| One parent foreign-born | 7.9 | 7.2 | 6.5 | 7.1 | 7.0 | 5.7 |
| Born in Finland with foreign-born parents | 2.3 | 2.0 | 1.5 | 2.1 | 2.0 | 1.5 |
| Born abroad with foreign-born parents | 3.0 | 2.5 | 3.2 | 4.1 | 2.6 | 3.5 |
| Urban-rural area (%) | n = 127,582 | n = 74,475 | n = 29,144 | n = 123,865 | n = 51,710 | n = 42,462 |
| Urban | 67.2 | 74.6 | 80.5 | 66.9 | 74.1 | 77.5 |
| Semi-urban | 18.5 | 14.8 | 13.7 | 18.5 | 15.7 | 17.7 |
| Rural | 14.3 | 10.6 | 5.8 | 14.7 | 10.2 | 4.8 |

school had at least one parent with higher educational degree. In upper secondary school the proportion was even higher, 65% in girls and 71% in boys. In vocational schools about a third of the students had a parent with higher educational degree. About 6–7% of the students had a foreign-born parent or were born abroad with one parent born in Finland. A smaller proportion were born in Finland (2%) or born abroad (2–4%) to foreign-born parents. About 66% of the students in lower secondary school, 74% in upper secondary school and 80% in vocational schools were in an urban area.

Table 2 shows the distributions of social connectedness in girls and boys in lower and upper secondary and vocational school. Between 45–51% of girls and 56–60% of boys reported having three or more close friends, while 6–7% of girls and 9–11% of boys had no close friends. About a third of girls and less than a quarter of boys felt sometimes lonely. Those quite often

**Table 2. Distributions of the social connectedness variables shown as % or mean and standard deviation (SD) in the School Health Promotion Study 2017–2021 in girls and boys in lower and upper secondary and vocational school.**

| Variable | Girls—Lower | Girls—Upper | Girls—Vocational | Boys—Lower | Boys—Upper | Boys—Vocational |
|---|---|---|---|---|---|---|
| Close friends (%) | n = 126,537 | n = 74,231 | n = 28,902 | n = 121,012 | n = 51,223 | n = 28,902 |
| None | 7.0 | 6.3 | 6.4 | 11.1 | 9.9 | 8.9 |
| One | 20.7 | 18.7 | 21.9 | 16.1 | 16.6 | 15.8 |
| Two | 24.3 | 23.5 | 26.3 | 16.1 | 17.2 | 15.1 |
| Three or more | 48.0 | 51.6 | 45.5 | 56.8 | 56.2 | 60.2 |
| Loneliness (%) | n = 126,729 | n = 74,284 | n = 28,936 | n = 121,388 | n = 51,319 | n = 28,936 |
| Never | 17.1 | 12.4 | 15.0 | 41.2 | 29.3 | 42.1 |
| Very seldom | 33.4 | 34.0 | 30.3 | 33.8 | 39.1 | 32.8 |
| Sometimes | 32.6 | 36.2 | 35.4 | 16.8 | 23.5 | 17.9 |
| Quite often | 12.4 | 13.4 | 14.3 | 4.4 | 6.1 | 4.8 |
| All the time | 4.5 | 4.0 | 5.0 | 2.9 | 2.1 | 2.4 |
| Belonging at school (mean, SD) | n = 125,664 | n = 74,092 | n = 28,584 | n = 119,492 | n = 51,115 | n = 40,972 |
| | 3.3 (1.0) | 3.4 (0.9) | 3.5 (0.9) | 3.7 (1.0) | 3.7 (0.9) | 3.8 (0.9) |

or all the time lonely was higher in girls (about 17%) compared to boys (about 8%). Sense of belonging at school was somewhat greater in boys and in vocational schools.

## 3.2. General trends in social connectedness between 2017 and 2021

Overall, the students reported having less often three or more close friends and more often no close friends or one close friend in 2021 compared to 2019 (S1 Fig in S2 File). This was equal to a drop for 56% to 50% of having three of more close friends and an increase from 7% to 9% of having no close friends. There was very little change between 2017 and 2019.

Loneliness increased overall (S2 Fig in S2 File), being about 13% higher in 2021 compared to 2017 (from score 2.17 to 2.46). This was mostly due to a smaller proportion of students reporting never lonely and a larger proportion reporting sometimes lonely, which further accelerated between 2019 and 2021 (S3 Fig in S2 File). Overall, the proportion of those never lonely dropped from 30% to 20% and the proportion of those sometimes lonely increased from 24% to 30% between 2017 and 2021. There was also some decline in the proportion of those very seldom lonely and an increase in the proportion of those who reported being sometimes, quite often or all the time lonely between 2019 and 2021.

Sense of belonging at school declined overall (about 8% from score 3.70 to 3.42), with a faster decline between 2017–2019 compared to the period between 2019 to 2021 (S4 Fig in S2 File). The decline was mostly due to the drop in the proportion of those who agreed (from 37% to 30%) or fully agreed (from 19% to 11%) with the statement that they felt to be an important member of class and school community (S5 Fig in S2 File). There was also an increase in the proportion of those who disagreed (from 8% to 12%) or neither agreed or disagreed (from 33% to 41%) with the statement that they felt to be an important member of class and school community.

## 3.3. The effects of gender and school level on the trends of social connectedness

**3.3.1. Number of close friends.**   Number of close friends declined from 2017 to 2021 (Table 3, Model 1 for number of close friends). Girls reported fewer close friends compared to boys. Students in lower secondary schools reported fewer close friends compared to the students in upper secondary or vocational schools. The three-way interaction year*gender*school level was not significant (Table 3, Model 1 for number of close friends). However, there were differences in trends of close friends between boys and girls (interaction year*gender, $p<0.001$) and between the school levels (interaction year*school level, $p<0.001$).

S6 Fig in S2 File shows that compared to boys, girls showed a faster increase of having no or one close friend and decrease of having three or more close friends. S7 Fig in S2 File shows that the differences in the trends of close friends by school levels were very small and mostly overlapping. However, there were initially (in 2017) slightly higher proportion of students in upper secondary and vocational school compared to lower secondary school students who had three or more close friends. Over the next two measurement occasions (2019 and 2021) the differences by school level disappeared.

**3.3.2. Loneliness.**   Loneliness increased between 2017 and 2021 (Table 3, Model 1 for loneliness). Girls reported more frequent feelings of loneliness compared to boys. Students in lower secondary schools reported loneliness more compared to the students in upper secondary or vocational schools. The three-way interaction year*gender*school level was significant, $p<0.001$ (Table 3, Model 1 for loneliness). S8 Fig in S2 File shows that loneliness increased in girls in all school levels and boys in upper secondary school through the study period 2017–2021. Boys in lower secondary and vocational schools did not show change in loneliness

**Table 3. Multivariable regressions for number of close friends, loneliness and belonging at school in secondary school students in Finland.**

| | Number of close friends [a] | | Loneliness [b] | | Belonging at school [c] | |
|---|---|---|---|---|---|---|
| | Model 1 | Model 2 | Model 1 | Model 2 | Model 1 | Model 2 |
| *Main effects* | | | | | | |
| Year of survey (ref = 2017) | | | | | | |
| 2019 | -0.03* | -0.01 | 0.00 | 0.02*** | -0.32*** | -0.31*** |
| 2021 | -0.10*** | -0.10*** | 0.11*** | 0.13*** | -0.34*** | -0.33*** |
| School level (ref = lower secondary school) | | | | | | |
| Upper secondary | 0.06*** | 0.03 | 0.09*** | 0.06*** | -0.13*** | -0.90*** |
| Vocational | 0.10*** | 0.07*** | 0.00 | 0.02*** | 0.12*** | 0.15*** |
| Female | -0.14*** | -0.18*** | 0.26*** | 0.27*** | -0.41*** | -0.36*** |
| Parent(s) with higher education degree | 0.14*** | 0.08*** | -0.01*** | -0.01*** | 0.09*** | 0.07*** |
| Immigration status (ref = Student and parents born in Finland) | | | | | | |
| One parent foreign-born | -0.15*** | -0.15*** | 0.05*** | 0.05*** | -0.06*** | -0.05*** |
| Born in Finland with foreign-born parents | -0.24*** | -0.24*** | -0.03*** | -0.03*** | 0.03** | -0.01 |
| Born abroad with foreign-born parents | -0.51*** | -0.51*** | 0.07*** | 0.06*** | -0.08*** | -0.14*** |
| Urban-rural (ref = urban) | | | | | | |
| Semi-urban | -0.03*** | -0.04*** | -0.01*** | -0.01 | 0.03*** | 0.03** |
| Rural | -0.09*** | -0.12*** | -0.00 | -0.01 | 0.06*** | 0.08*** |
| *Wald tests for interactions*: | Wald (df) | Wald (df) | Wald (df) | Wald (df) | Wald (df) | Wald (df) |
| Year*gender*school level | - | - | 19.13 (4)*** | - | 23.12 (4)*** | - |
| Year*gender | 201.59 (2)*** | - | - | - | - | - |
| Year*school level | 23.63 (4)*** | - | - | - | - | - |
| Year*gender*parental education | | 7.22 (2)* | | - | | 14.90 (2)*** |
| Year*gender*immigration | | - | | - | | - |
| Year*gender*urban-rural | | - | | - | | - |
| Year*school level*parental education | | 9.99 (4)* | | - | | - |
| Year*school level*immigration | | - | | - | | - |
| Year*school level*urban-rural | | - | | 18.09 (8)* | | 52.42 (8)*** |
| Year*parental education | | - | | - | | - |
| Year*immigration | | - | | - | | 24.88*** |
| Year* urban-rural | | 11.92 (4)* | | - | | - |

Model 1: interactions for year*gender* school level only; Model 2: interactions for year*gender* school level + parental education, immigration or urban-rural indicator.

Notice: Unstandardized estimates for the final model shown; none of the four-way interactions were significant in the final model (Year*gender*school level*immigration, Year*gender*school level*immigration, Year*gender*school level*urban-rural); higher-order non-significant interactions removed from the model shown; see S1 File for the estimates and Wald testing at each step.

[a] Ordered logistic regression, $n = 415,932$;

[b] Generalized linear model, $n = 416,572$;

[c] Generalized linear model, $n = 415,367$; df = degrees of freedom.

*** $p < 0.001$,

** $p < 0.01$,

* $p < 0.05$

between 2017–2019, but during the latter part of the study period (2019–2021) their loneliness also increased. Boys in upper secondary school showed higher loneliness compared to the boys in lower secondary or vocational schools. In girls, the differences in loneliness by school level were smaller, and the highest levels of loneliness were in upper secondary and vocational schools.

**3.3.3. Belonging at school.** Sense of belonging at school declined from 2017 to 2021 (Table 3, Model 1 for belonging at school). Girls reported less school belongingness compared to boys. Students in lower secondary schools reported more school belongingness compared to the students in upper secondary, but less school belongingness compared to the students in vocational schools. The three-way interaction year*gender*school level on sense of belonging at school was significant, $p<0.001$ (Table 3, Model 1 for belonging at school). The sense of belonging at school were overall highest in vocational school (S9 Fig in S2 File) and lowest in upper secondary school in 2017 and in lower secondary schools in 2019 and 2021. There was a faster decline in school belongingness in boys in lower secondary school compared to their female counterparts between 2017 and 2019. However, girls in lower secondary school continued to show a decline in their school belongingness between 2019 and 2021.

## 3.4. The effects of socioeconomic factors on the trends of social connectedness

Generally, a higher parental degree was associated with reports of more close friends, less loneliness and more belongingness at school (Table 3, Model 1 for number of close friends, loneliness and belonging at school). Students born in Finland with Finnish-born parents (no immigration background) reported more close friends compared to the students in other immigrant groups. Students without immigration background also reported less loneliness and more belongingness at school compared to the students with one foreign-born parent or the students born abroad to foreign-born parents. The students born in Finland to foreign born parents reported less loneliness and more belonging at school compared to the students without immigration background. The students in urban schools reported more close friends and less belonging at school compared to the students in semi-urban or rural schools. The students in semi-urban schools reported less loneliness compared to the students in urban or rural schools.

We tested the interactions by parental education, immigration status of the student and urban/rural location of the school on each of the three social connectedness variable while taking into account the relevant variation by year, gender and school level (year*gender*school level).

**3.4.1. Number of close friends.** None of the four-way interactions of parental degree, immigration status of the student or urban-rural location of the school by year*gender*school level was significant (Table 3, Model 2 number of close friends). However, there were slight differences in trends of close friends between boys and girls by parental degree (the interaction year*gender*parental degree, $p<0.05$) and between the school levels by parental degree (the interaction year*school level*parental degree, $p<0.05$). Number of close friends also differed between urban-rural location of the school (the interaction year*urban-rural location of the school, $p<0.05$).

Girls whose parents had no degree reported fewer close friends, whereas in boys the parental degree made less difference in the number of close friends, especially in 2019 (apart from the lower frequency of three of more close friends in those with no parental degree) (S10 Fig in S2 File). Upper secondary school students whose parents had no degree reported fewer close friends, whereas for the students in lower secondary and vocational school the parental degree made less difference in the number of close friends, especially in 2019 (apart from the lower frequency of three of more close friends in those with no parental degree) (S11 Fig in S2 File). Although the students in urban schools reported more often having three or more close friends compared to the students in rural schools in 2017, this difference by the locations of the school

disappeared by 2021 because of the faster decline in having three or more close friends in urban schools (S12 Fig in S2 File).

**3.4.2. Loneliness.**   None of the four-way interactions of parental degree, immigration status of the student or urban-rural location of the school by year*gender*school level was significant (Table 3, Model 2 for loneliness). There were slight differences in the trends of loneliness between the school levels by urban-rural location of the school (the interaction year*school level*urban-rural location, $p<0.05$).

The differences in loneliness by the school's urban-rural location were small in lower secondary schools throughout the study period. Loneliness in vocational schools appeared to increase faster in urban locations (no differences by location in 2017) (S13 Fig in S2 File). In upper secondary school, the increase in loneliness was faster in urban locations between 2017 and 2019, but the differences by locations were small in 2021.

**3.4.3. Belonging at school.**   None of the four-way interactions of parental degree, immigration status of the student or urban-rural location of the school by year*gender*school level was significant (Table 3, Model 2 for belonging at school). However, there were differences in trends of belonging at school between boys and girls by parental degree (the interaction year*gender*parental degree, $p<0.001$). School level also interacted with urban-rural location of the school (the interaction year*school level*urban-rural location, $p<0.001$). There was also an interaction between year and immigrations status of the student (the interaction year*immigration status, $p<0.01$).

The decline in the sense of belonging at school were faster in girls whose parents did not have a degree compared to girls with parental degree (S14 Fig in S2 File). In boys the differences by the parental degree remained quite similar over the study period.

Sense of belonging at school were lowest in students born abroad with foreign-born parents in 2017 (S15 Fig 15 in S2 File). However, the sense of belonging at school declined sharply between 2017 and 2019 in the students without immigration background and those with one foreign-born parent so that there were little differences between them by 2021. Similar to the students born aboard with foreign-born parents, the students with one foreign-born parent showed less decline over time, especially between 2019–2021, and the former mentioned group reported higher belonginess compared to all other groups in 2021.

In lower and upper secondary school, the sense of belonging at school was highest in rural schools, but the differences by the location mostly disappeared in 2019 in lower secondary school and 2021 in upper secondary school (S16 Fig in S2 File). In vocational schools there were no differences in the sense of belonging at school by urban-rural location in 2017 but in the latter two time points (2019 and 2021), belongingness was greater in rural schools, i.e. the decline was slower, especially between 2017 and 2019, in rural schools compared to urban and semi-urban schools.

S1 File shows the odds ratios and exponentiated betas with average marginal effects for the final models. The effect sizes of the covariates on the outcomes were mostly small to moderate (see the interpretation in S1 File). The largest effects were related to gender differences and change over time in social connectedness.

## 4. Discussion

The current study explored the cohort trends of social connectedness (having close friends, feelings of loneliness and sense of belonging at school) in secondary schools in Finland between 2017 and 2021. It also investigated whether these trends vary by gender, school level, parental education, immigration status of the student and urban-rural location of the school. The results illustrate a general declining trend of social connectedness in the cohorts of

secondary school students in Finland between 2017 and 2021. These trends were however not uniform across all the students, but varied by gender, school level and sociodemographic background (parental education, immigrations status and urban or rural location). The results concerning the three research questions are discussed in detail below.

## 4.1. The general trends of social connectedness

Social connectedness declined in the three consecutive cohorts of secondary school students in Finland between 2017 and 2021. Along with gender differences, declining trends were the most prominent features of the results (small to moderate effect sizes; [64, 65]. The results are in line with the previous observations of increasing loneliness and decreasing sense of school belonging up to 2018 [5, 19, 21, 22], and the accelerated rates of social isolation, loneliness and decreased sense of connectedness during pandemic [25, 26].

Decline in sense of belonging at school was mostly in the former part of the study period (2017–2019), although there was further but not as fast decrease between 2019 and 2021. Increase in loneliness and decline in the number of close friends was almost entirely in the latter part of the study period (2019–2021, i.e. before and after the pandemic). The results suggest that the decline in the sense of school belonging had already started well before the pandemic started. Pandemic appeared to coincidence especially with the decline of social connectedness in those who reported having three or more close friends and being never lonely in 2019. This suggest that the social restrictions might have hit hardest those with most social connections. These groups showed some decline already between 2017 and 2019, whereas those with less than three close friends and being lonely very seldom or more often hardly changed between 2017 and 2019. A differential trend in the social domains have been reported earlier. In a Swedish study covering an earlier period (2000–2018), the findings showed that in the summary score for school belonging the decline was mostly accounted for the items measuring general orientation to school such as feeling outsider and feeling belonging and less for the items focusing on peer relationships at school (e.g. making friends easily) [22].

## 4.2. The trends of social connectedness by gender and school level

Girls reported fewer close friends, greater loneliness and sense of not belonging at school compared to boys. The decline in close friends and increase in loneliness was faster in girls compared to boys between 2017–2019, except for boys in upper secondary school whose pattern of change was similar to girls. Both boys and girls showed increase in loneliness and decline in having close friends between 2019 and 2021. The results are congruent with the previous findings on increasing trends in loneliness in both boys and girls in upper secondary school in Norway between 2014–2018 [51], and the increase in loneliness especially among girls after 2015 in a large multinational study between 2000–2018 [5] and consistently higher level and faster increasing loneliness in adolescent girls compared to boys between 2006 and 2018 [19]. The current study extends the time after pandemic and suggests that social connectedness had declined further between 2019–2021, regardless of gender. They are in line with consistently higher loneliness in girls compared to boys before and during the pandemic [20, 28].

The current study found that boys in all school levels expressed greater sense of belonging at school compared to girls. This gender difference has also been found previously [22], but some others have reported girls having a greater sense of school belonging [48]. The different findings may be due to the different location of the study and study period being later (after 2015) than what was covered in the previous study [48]. The faster decline in school belongingness appeared to happen between 2017–2019, especially in boys in lower secondary school.

Girls and students in upper secondary school showed a continuing decline in school belonging-
ingness over time, which accelerated between 2019–2021.

In girls and upper secondary school students, the feelings of being connected to school
might have been especially at risk due to the pandemic restrictions, as these groups often show
higher educational aspirations and prepare for the competition to enter higher education [66].
Moreover, these groups were also affected by the national changes in university entrance
requirements from 2017 onwards which shifted the focus from entrance examinations to ear-
lier performance in the secondary school [67]. In a previous Swedish study, the decline in
school belonging appeared to coincidence with major education reforms [22], which included
standardised and stricter nationwide curriculum, testing and grading leading to a more perfor-
mance-oriented learning environment.

Another interesting finding was the consistently higher sense of school belonging in the stu-
dents in vocational school compared to those in lower or upper secondary school. This sug-
gests that the students in vocational schools may experience a better fit into the school
environment compared to their fellow students in lower or upper secondary schools, challeng-
ing the idea of better school connectedness in academic vs. vocational tracks [52, 54]. It is note-
worthy that moving between the academic and vocational tracks is flexible in the Finnish
school system. Choosing a vocational education at the end of the basic education (grade 9)
does not close the doors to enter higher education after completing a vocational qualification
[59]. It is important to note that the sample for the students in vocational school was smaller
compared to the other two school levels and the participation rates varied by the fields of stud-
ies in vocational school. Therefore, the sample of vocational school students may not be repre-
sentative of all students in vocational secondary schools in Finland.

## 4.3. The trends of social connectedness by parental education, immigration status of the student and urban-rural location of the school

Some results showed lower levels of social connectedness in students with lower parental edu-
cational level and immigration background. This is in line with other findings on the differ-
ences by socioeconomic and immigrations status [21, 22]. However, when looking at the
subgroups of students the effects were not uniform: parental higher education was associated
with having more close friends in girls and in upper secondary schools in all three time points.
Although a similar but smaller difference by parental degree was observed in boys and in
lower secondary or vocational schools in 2017, there were few differences after that. Moreover,
in girls, the sense of belonging at school declined faster in those with no parental higher degree
compared to those with parental higher degree. In boys the differences by parental education
remained similar across the time points.

The results mentioned above suggest the possible protective role of parental educations in
these groups. As discussed above, girls and upper secondary school students show higher edu-
cational aspirations [66] and their socioemotional wellbeing may be more susceptible to insti-
tutional changes (such as changes in university entrance requirements and pandemic).
Parental education may play a part in mitigating the stress and enhance social interactions in
these groups of students. On the other hand, restricting the access to social spaces such as
school and work may have a bigger impact on the social life of girls and women compared to
boys and men. When confined at home, as happened during the lockdowns, girls and women
may more easily adopt traditional caring roles within the family at the cost of keeping contact
with friends or making bonds in the school's social circles [31]. These gender-related social
dynamics may be exacerbated by other underlying vulnerabilities such as a lower socioeco-
nomic status which itself was likely to deteriorate due to the economic downturn in the

Western countries, including Finland [68]. It is notable that the economic recession was well on the way [69] before the harmful economic effects of the pandemic lockdowns [29, 31]. This intersection of gender and socioeconomic status in the challenging times may explain the consistently lower and often faster decline of social connectedness in girls with lower versus higher parental education compared to a higher social connectedness in boys with a smaller impact of parental education.

The levels of social connectedness also varied by urban-rural locations of the school depending on the type of social connectedness measure and time. Students in rural lower and upper secondary schools reported initially a greater sense of belonging at school compared to the students in semi-urban and urban schools. Students in rural schools initially had less often three or more close friends compared to their counterparts in urban schools. These differences by location however disappeared by 2021. In vocational schools, rural location was associated with less loneliness and greater school belongingness. The results suggest that urban environment might have initially been beneficial for having more close friends, whereas rural locations of the school in some cases would facilitate sense of belonging at school and feeling less lonely [42–44].

Over time the differences in social connectedness by urban-rural location of the school tended to diminish (fan in), as described above. A similar pattern was seen by immigration status: First-generation immigrants initially reported less belonging at school compared to the other groups of students with or without immigration background. However, by 2019 the differences between the immigration groups decreased or disappeared because of a faster decline in feelings of belonging at school especially in students with partial or no immigration background compared to first- and second-generation immigrants. The results for declining differences by immigration status and urban-rural location of the school are in line with the findings in a Danish data between 1991–2014. The Danish study showed decreasing social inequality in loneliness because of those with more advantageous background getting lonelier [43]. Högberg et al. [22] reported that although school belongingness declined most in foreign-born students, students from disadvantaged background and low achieving students, the changes in background demographic factors or school environment were not associated with the decline.

The possible reasons for declining social connectedness and narrowing social gradient might be larger macro-level changes that may have had more impact on those with more advantageous positions. For instance, increasing use of social media and smartphones, declining wellbeing and economic trends have been cited [5] as well as school reforms [22]. These institutional and societal events may hit those with the least advantaged socioeconomic position and least resources hardest [22] or affect more those families who have more socioeconomic resources and higher expectations of the future opportunities [43]. Social restrictions during the pandemic reduced the opportunities for participation in activity groups. As many of them charge participation fees, the closures were likely to affect the social interactions in those families who could afford them, i.e. those with more socioeconomic resources.

An alternative interpretation might be that the reduced feelings of belonging, in this case 'school belonging', reflects a wider shift in experiencing belonging across social, national, and cultural borders [16], or being 'at-home' in the present society [17, 18]. From 2017 to 2021 the students viewed themselves less often as an important member of the classroom and school community. The decreased sense of belonging at school was particularly evident in the large majority. There was less change in school belongingness in the minority groups, such as immigrant students. It is noteworthy that these shifts coincidence with a general decline in school belongingness in many OECD countries [21] and precede the further smaller decline between 2019 and 2021 which could be attributed to the pandemic restrictions and changes in social

interactions. Regardless of these observations in the previous studies, there has been very little attempt to unravel the causes for the declining overall trends. The declined sense of belonging at school has been mostly discussed as a microsystem problem concerning the learner, peers, teacher and sometimes home environment but rarely as a part of wider systems or as a political question [17, 70]. In the recent years, a prominent political interest in Finland and globally has been in inequity, e.g. concern on the widening gaps between sub-populations, intense focus on the vulnerable groups and minorities, and most recently, associating pandemic with inequalities [33, 71]. The overall message is that the disadvantaged fair worse and decline faster in various measures of wellbeing. This has been followed by further calls for equity and collective efforts to tackle inequalities, e.g. urging to step aside from individualistic position and embrace the values and practices of sharing and solidarity, see [72]. Meanwhile the concurrent school budget cuts in Finland have had a detrimental effect on everyone, regardless of background [73]. In the light of the present results, the question is whether the decreased sense of belonging in the majority was a response to have been sidelined in the discussion, not feeling 'at-home' or important anymore. In Finland's case, the rapid shift from praising the success of equality in school in the early 2010s [74, 75] to blaming the system being failed and unfair in the latter part of the 2010s [76] may not have gone unnoticed by the school communities and may have created feelings of division and alienation, both in teachers and students.

Whatever the reason, ignoring a major cohort decline in sense of belonging can have a high price if increasing numbers of young people in the future cannot feel being part of their social groups, consequently leading to feelings of alienation. In 2021, the most frequent response to the question of whether the student felt being an important member of school was that they neither agreed nor disagreed (41%), suggesting feelings of indifference. It is important to investigate whether this stance has the same or even worse implications for social connectedness as disagreeing with the statement. It is also important to find out whether the trends of declining sense of belonging at school—along with having fewer close friends and increased loneliness—continue in the future and whether sense of belonging at school extends to other areas of life, such as being a member of family, work team and society.

### 4.4. Limitations of the study

Although the study includes a large representative sample of secondary school students in Finland, the large-scale survey focuses on health-related topics and has a limited set of explanatory background variables and measures of social connectedness. Educational level of the parent and immigration status were based on the students' self-reports, which may contain reporting error [58]. As the students filled in the survey by themselves, the participation required sufficient skills in Finnish or Swedish (two official languages in Finland) or any of the three other languages the questionnaire was translated into (Russian, English, North Sami). The questionnaire was also available in plain Finnish and Swedish. Despite these options, it is possible that some newly immigrated students or other students with language barriers were not included. This may impact the representativeness of the sample in some of the subgroups and may over- or underestimate the levels of social connectedness in these groups. Because of the selected measures used, some of the findings may be difficult to compare with other studies using a more comprehensive or different measure for social connectedness or individual and environmental background factors. The future data linkage to school-specific characteristics will make it possible to study the sources of school-level variation in addition to the fixed individual and school characteristics.

The participation rate in vocational schools varied by the field of study and was lower than in lower and upper secondary schools. In vocational schools, only students under 21 years

were included making it challenging to reach and motivate the students in the target group to participate. Moreover, some students were doing their obligatory work placements as part of the vocational qualification and were not available at the time of the data collection. It is possible that some of the differences were due to the selection of the vocational school students in the study. It is however noteworthy that all the models were adjusted for the key background factors, and therefore any differences because of them were taken into account in the models. The study described the cross-sectional cohort trends over three measurement occasions. It does not provide evidence of the longitudinal change within individuals. It also does not contain a control group, so the changes between 2019–21 may be attributed to the pandemic or other events in the time period.

## 5. Conclusions

The results suggest an overall decline in social connectedness in secondary school students. The adverse trends in social connectedness are often associated with female gender and upper secondary school. However, the results do not suggest widening socioeconomic differences. On contrary, the differences, if there are any, remain (parental education) or the more advantaged groups decline faster, i.e. move towards the less advantaged groups (immigration status of the student and urban-rural location of school) by 2021.

These key findings are important in at least three ways: first, the current study with using a representative national sample of students with same measurements before and across the pandemic is valuable as it overcomes some of the common problems of accessing reliable and representative data during covid-19 [77].

Second, the current findings add to the evidence base of the recent negative trends in psychosocial wellbeing in young people [25, 26, 33, 37, 38]. Considering these known negative effects of eroding social networks and loneliness on wellbeing and development of young people, the recent overwhelming evidence of the negative impact of covid-19 measures at schools on social isolation and sense of connectedness and further mental health issues should not become as a surprise. The restrictions were directly targeting social connections while it is well-established that these connections are vital for young people's development and wellbeing [3, 29, 78–80]. The previous research points that social connections in adolescence have more long-term associations with mental health in the later life than for instance academic achievement [81]. Therefore, the declining trends of social connectedness should not be overlooked. Although there has been an ongoing discussion in Finland on the best ways to protect children's rights to education after the pandemic, surprisingly little has been said about the rights to social connectedness. The views, especially at the early stages of the pandemic, seemed to back the move to online learning and the benefits of it [82, 83]. However, there were also concerns on the impact of school closures to loneliness and health of the students [27]. Finnish policy makers acknowledge the significance of the pandemic on young peoples' wellbeing and learning and that it will take several years to address the negative consequences [84]. Similar to many other countries [85, 86], the proposed actions are general, not specified at a detailed level, relying on further commissioned research and postponed to the future.

Third, the current study shows the adverse trends not only in the disadvantaged groups but even more prominently in the large majority of the so-called 'mainstream' secondary school students. The increased equality, the narrowing gap, was gained at the expense of social wellbeing of many. The scale of the impact is extensive, touching many young people in these cohorts. This means that the narrow and compartmentalised focus on inequities [33–35], which can potentially create more division, should be widened to cover practically everyone and many other key areas in young people's life than only academic achievement and

immediate wellbeing. The approach needs to be suitable for each local context and respect young people's needs and rights.

Feelings of social outsiderhood pose serious challenges for young people's mental and physical wellbeing, increase social inequality and enforce further erosion of social capital in society [2, 3, 6]. Therefore, the promotion of child and youth well-being through decreasing loneliness and strengthening sense of belonging to learning groups and school is a cross-cutting priority at different levels of society. To increase social connections and feelings of belonging at school, we need practical solutions and recommendations to changes in the national core curricula, education practices, students' wellbeing related services, interprofessional training, service integration and public policy-related formulations. Listening to young peoples' own voice in these changes is crucial.

## Supporting information

**S1 File. Stata outputs for the models.**
(PDF)

**S2 File. Supplementary figures.**
(PDF)

**S3 File. Postestimation diagnostics and missing data patterns.**
(PDF)

## Author Contributions

**Conceptualization:** Sanna Read, Katariina Salmela-Aro.

**Data curation:** Jenni Helenius.

**Formal analysis:** Sanna Read.

**Funding acquisition:** Katariina Salmela-Aro, Niina Junttila.

**Investigation:** Sanna Read, Jenni Helenius.

**Methodology:** Sanna Read, Noona Kiuru.

**Project administration:** Katariina Salmela-Aro, Niina Junttila.

**Resources:** Katariina Salmela-Aro, Jenni Helenius.

**Software:** Sanna Read.

**Supervision:** Katariina Salmela-Aro.

**Validation:** Sanna Read.

**Visualization:** Sanna Read.

**Writing – original draft:** Sanna Read.

**Writing – review & editing:** Sanna Read, Katariina Salmela-Aro, Noona Kiuru, Jenni Helenius, Niina Junttila.

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
