## [Decision Letter · Decision Letter 0]

18 Jul 2024

PONE-D-24-04572The cohort trends of social connectedness in secondary school students in Finland between 2017 and 2021PLOS ONE

Dear Dr. Read,

Thank you for submitting your manuscript to PLOS ONE. After careful consideration, we feel that it has merit but does not fully meet PLOS ONE’s publication criteria as it currently stands. Therefore, we invite you to submit a revised version of the manuscript that addresses the points raised during the review process.

We look forward to receiving your revised manuscript.

Kind regards,

Asami Shinohara

Academic Editor

PLOS ONE

Additional Editor Comments:

Thank you for considering your submission to PLOS ONE. First, I apologize for the delay in finding reviewers and making a decision.

Two reviewers, familiar with this field, have read your manuscript and provided valuable comments. Both reviewers found your research interesting and important but believe that some minor revisions need to be made. Please review the reviewers' comments and revise your manuscript accordingly.

As an Editor, I would like you to ensure that your manuscript meets the STROBE checklist (http://www.strobe-statement.org). For instance, please describe the SHP data in more detail, including the period and location of data collection, the total number of participants in SHP, and whether this is a follow-up survey or not. Also, if you do not use the entire SHP data from 2017-2021, please describe the eligibility criteria. In the results section, if this is a follow-up survey, please include information about how many participants joined this survey throughout 2017 to 2021. Additionally, please include 95% confidence intervals in the table.

We look forward to receiving your revised manuscript.

Reviewers' comments:

Reviewer's Responses to Questions

**Comments to the Author**

1. Is the manuscript technically sound, and do the data support the conclusions?

Reviewer #1: Yes

Reviewer #2: Yes

2. Has the statistical analysis been performed appropriately and rigorously? 

Reviewer #1: Yes

Reviewer #2: Yes

3. Have the authors made all data underlying the findings in their manuscript fully available?

Reviewer #1: Yes

Reviewer #2: No

4. Is the manuscript presented in an intelligible fashion and written in standard English?

Reviewer #1: Yes

Reviewer #2: Yes

5. Review Comments to the Author

Reviewer #1: Thank you for the opportunity to review this paper. The article details an interesting and timely study focusing on trends of social connectedness in secondary schools in Finland, with measures regarding number of close friends, feelings of loneliness, and sense of belonging. It outlines notable findings on trends of social connectedness relating to gender and sociodemographic context/background.

I find the paper to be worthy of publication, with some minor revisions. My suggestions for revisions are as follows:

1. There is an in-depth literature review of studies that explore social connectedness in adolescents, and it is explained that social connectedness is not a uniform concept and is a term subject to debate (page 3). However, there could be further recognition of the debates surrounding the concept of ‘belonging’ and ‘sense of belonging’. I think these arguments are important to tease out, given that conceptualisations of belonging have at times been under theorised and vaguely defined, according to Antonsich (2010) for example. Some scholars have theorised belonging as embedded in ‘multiple constellations across social, national, and cultural borders’ (Röttger-Rössler, 2018). It has also been conceptualised as a feeling of being ‘at-home’ (Antonsich, 2010) and as contrasted against feelings of ‘non-belonging’ (Anthias, 2016). How do such debates in the literature relate to this study’s rationale, methodology, and findings?

2. Further elucidation of the authors’ understanding of belonging may also help to address my second point, which concerns the measures of sense of belonging. The items from the questionnaire concern young people’s view of themselves as an ‘important member’ of the classroom community and school community (page 7). How does this sense of importance translate to a sense of belonging? I believe further clarification in the introduction of the authors' definition of belonging, as connected to these measures, would help to strengthen the arguments in this paper.

3. The participation rates for vocational schools were much lower (30%) than that for lower secondary and upper secondary school. The implications of this for the results are discussed in the Limitations section (page 21), however I wondered if there is any reason why participation rates were so much lower? Is there some contextual information regarding vocational schools which may be pertinent here?

4. In the Introduction and Discussion it is highlighted that social connectedness and social isolation in young people have become more visible concerns in the aftermath of the COVID-19 pandemic and lockdown. I wondered if there had been any attempts to address these concerns in Finnish educational policy or in political discourse in the country? Providing this context may point towards further implications for this study e.g. the need for developments in policy and practice to address these worrying trends in decreasing social connectedness.

Overall, I believe this is a strong paper and further engagement with the literature around belonging will help to clarify the methodology and strengthen the findings and conclusions.

Anthias, F. (2016). Interconnecting boundaries of identity and belonging and hierarchy-making within transnational mobility studies: Framing inequalities. Current Sociology, 64(2), 172-190.

Antonsich, M. (2010). Searching for belonging–an analytical framework. Geography compass, 4(6), 644-659.

Röttger-Rössler, B. (2018). Multiple Belongings. On the Affective Dimensions of Migration. Zeitschrift für Ethnologie, (H. 2), 237-262.

Reviewer #2: This paper by Sanna and colleagues examined trends in adolescent students’ social connectedness using cross-section surveys in Finland. Several regression models were estimated adjusted for relevant socio-demographic covariates. I have not much to say about this paper, other than mention that it is robust both theoretically and methodologically. I just have a few burning questions and suggestions that I outline below.

1. It would be very helpful to the reader to state more clearly what the purpose of the study is at the very beginning paragraph.

2. The article would benefit by explaining the context of Finland more. What do we know already about loneliness and connectedness in Finland?

3. At some point in the introduction, loneliness is treated as the opposite of social connectedness, i.e., “Apart from the recent social restrictions related to the pandemic, other possible reasons for the trends of increasing loneliness and declining social connectedness” (page 3). Perhaps, there needs to be a clearer theoretical conceptualisation of social connectedness?

4. I am wondering why age/grade was not included in the regression models considering that there are significant developmental changes occurring in this period? I do not think that there is a good rationale against including age in the models.

5. The article takes a multidimension approach to measuring social connectedness. Even though there is theoretical support for choosing these three indicators of social connectedness (i.e., loneliness, number of friends, belonging), I would like to invite the authors to show empirically using a principal components analysis/ exploratory factor analysis that these three indicators can actually measure a single concept of social connectedness. This is particularly important since the authors state themselves (page 3, line 44) that the measurement of social connectedness is debated.

6. The authors mention the use of robust standard errors but I am uncertain whether the regression models account for the clustering of the students within classrooms and schools as well. What was the intra-class correlation coefficient for each sampling level?

7. Following up to my comment above, I would also appreciate a clearer description of the sampling design (e.g., detailed explanation of stratification, clustering, randomisation, if applicable).

8. It would be great if the authors could provide the Little’s MCAR test to and examine the missing values’ patterns as well. Listwise deletion is unlikely to affect the estimates in such a big sample, but for the sake of thoroughness, I would advise considering this.

9. Ordered logistic regression assumes proportional odds/ parallel regression. Also, there are other regression assumptions, such as residual normality or the absence of multicollinearity. Meeting assumptions has not been mentioned inside the manuscript, but it is important to verify that the model is robust to these before proceeding with the key results.

10. There is a good discussion of the findings in light of previous evidence. Meanwhile, I do certainly feel that the discussion section begins rather abruptly (lines 375-379). The readers would appreciate a smoother recap of the aims of the study, the context a bit, the strengths and the key contributions.

11. What is missing from the discussion section is a brief paragraph/subsection on the policy and practice implications of the findings.

At this point I would like to invite the authors to proofread the paper once more for clarity and conciseness. For example, “Social connectedness is nevertheless not a uniform concept”. I am not certain that the term ‘uniform concept’ is clearly explaining what the authors wish here. “during school years” during the school years (page 3). This sentence is hard to follow because the syntax is confused: “For instance, Högberg et al. [18] reported that although school belongingness declined most in foreign-born students, students from disadvantaged background and low achieving students in Sweden between 200-2018, initially in 2000 there were little differences in school belongingness by these background characteristics.” Another typo: “Because of the selected measured used”

I do not have the time to do a proofreading throughout the manuscript but please take some time to review the language as well.

Overall, a very good population-based study showing declining trends in social connectedness. I just feel that the authors need to iron out the issues I raised above.

Kind regards

6. PLOS authors have the option to publish the peer review history of their article (what does this mean?). If published, this will include your full peer review and any attached files.

Reviewer #1: **Yes: **Jonathan Hancock

Reviewer #2: No

---

## [Author Response · Author response to Decision Letter 0]

5 Sep 2024

Response to Reviewers

Thank you for your thorough and very helpful reviews. We have amended the text to cover each point and hope that after these improvements it can be accepted. Note: the added text is in red in the revised manuscript and our responses can be found after each comment below. 

Reviewer #1: Thank you for the opportunity to review this paper. The article details an interesting and timely study focusing on trends of social connectedness in secondary schools in Finland, with measures regarding number of close friends, feelings of loneliness, and sense of belonging. It outlines notable findings on trends of social connectedness relating to gender and sociodemographic context/background.

I find the paper to be worthy of publication, with some minor revisions. My suggestions for revisions are as follows:

1. There is an in-depth literature review of studies that explore social connectedness in adolescents, and it is explained that social connectedness is not a uniform concept and is a term subject to debate (page 3). However, there could be further recognition of the debates surrounding the concept of ‘belonging’ and ‘sense of belonging’. I think these arguments are important to tease out, given that conceptualisations of belonging have at times been under theorised and vaguely defined, according to Antonsich (2010) for example. Some scholars have theorised belonging as embedded in ‘multiple constellations across social, national, and cultural borders’ (Röttger-Rössler, 2018). It has also been conceptualised as a feeling of being ‘at-home’ (Antonsich, 2010) and as contrasted against feelings of ‘non-belonging’ (Anthias, 2016). How do such debates in the literature relate to this study’s rationale, methodology, and findings?

-Thank you for bringing up this important perspective. Although our sense of belonging measure only focuses on school environment, the expanded definition is useful in conceptualising the study and interpreting the results. We have added the key concepts and references suggested above to the introduction (p.3) and discussed the findings from this perspective later in the text (pp. 22-23). 

2. Further elucidation of the authors’ understanding of belonging may also help to address my second point, which concerns the measures of sense of belonging. The items from the questionnaire concern young people’s view of themselves as an ‘important member’ of the classroom community and school community (page 7). How does this sense of importance translate to a sense of belonging? I believe further clarification in the introduction of the authors' definition of belonging, as connected to these measures, would help to strengthen the arguments in this paper.

-Many thanks for the pointing this out. As mentioned above, we have added more background of sense of belonging and discuss it from this point of view, taking into account the wording in the measure. As the current study investigates belonging at class and school community and the two questions were formulated to measure this, the focus is on the sense of belonging as an important or dear group member in this context. 

3. The participation rates for vocational schools were much lower (30%) than that for lower secondary and upper secondary school. The implications of this for the results are discussed in the Limitations section (page 21), however I wondered if there is any reason why participation rates were so much lower? Is there some contextual information regarding vocational schools which may be pertinent here?

-Thank you for the comment. The participation rates in vocational schools have been lower than in lower and upper secondary schools in each data collection since the first data collection in vocational schools in 2008. The main reasons for the lower response rates are that vocational schools do not provide students with the opportunity to respond as systematically as other schools. Some students were doing work placements at the time of the survey. In vocational schools, a large proportion of students are over 21 years old, meaning they were not part of the target group. Therefore, reaching and motivating those under 21 to participate was more difficult in vocational schools than at other educational levels. We have added information on these factors in the limitations section when we discuss the lower participation rate in vocational schools (p. 24).

4. In the Introduction and Discussion it is highlighted that social connectedness and social isolation in young people have become more visible concerns in the aftermath of the COVID-19 pandemic and lockdown. I wondered if there had been any attempts to address these concerns in Finnish educational policy or in political discourse in the country? Providing this context may point towards further implications for this study e.g. the need for developments in policy and practice to address these worrying trends in decreasing social connectedness.

-Thank you. We have added more text on these in the conclusions section, p. 25.

Overall, I believe this is a strong paper and further engagement with the literature around belonging will help to clarify the methodology and strengthen the findings and conclusions.

Anthias, F. (2016). Interconnecting boundaries of identity and belonging and hierarchy-making within transnational mobility studies: Framing inequalities. Current Sociology, 64(2), 172-190.

Antonsich, M. (2010). Searching for belonging–an analytical framework. Geography compass, 4(6), 644-659.

Röttger-Rössler, B. (2018). Multiple Belongings. On the Affective Dimensions of Migration. Zeitschrift für Ethnologie, (H. 2), 237-262.

Reviewer #2: This paper by Sanna and colleagues examined trends in adolescent students’ social connectedness using cross-section surveys in Finland. Several regression models were estimated adjusted for relevant socio-demographic covariates. I have not much to say about this paper, other than mention that it is robust both theoretically and methodologically. I just have a few burning questions and suggestions that I outline below.

1. It would be very helpful to the reader to state more clearly what the purpose of the study is at the very beginning paragraph.

-Thank you for the comment. We have added the purpose of the study at the end of the first paragraph (p. 3). 

2. The article would benefit by explaining the context of Finland more. What do we know already about loneliness and connectedness in Finland?

-Thank you for this comment. We have included more information on the levels of loneliness and school belonging in Finland from the same period, see introduction (p. 4-5). There are only few cohort trend studies and therefore the existing information relies mostly on single time points. 

3. At some point in the introduction, loneliness is treated as the opposite of social connectedness, i.e., “Apart from the recent social restrictions related to the pandemic, other possible reasons for the trends of increasing loneliness and declining social connectedness” (page 3). Perhaps, there needs to be a clearer theoretical conceptualisation of social connectedness?

-Thank you for this comment. We have modified the sentence in question to refer to the specific findings we discuss in the previous paragraphs: “Apart from the recent social restrictions related to the pandemic, other possible reasons for the above mentioned trends of increasing loneliness and declining sense of belonging…” . We have also clarified the terms used in the study in the introduction and come back to these in discussion, see also the response to Reviewer 1, comment 1. 

4. I am wondering why age/grade was not included in the regression models considering that there are significant developmental changes occurring in this period? I do not think that there is a good rationale against including age in the models.

-Thank you. As the current study uses three school levels which are age-specific, adding age to the model would produce a nearly complete overlap with school levels. Age differences are also very little in lower and secondary schools as they only include year 8 and 9 students in lower secondary and first- and second-year students in upper secondary. There is some age variation in vocational schools, but to include this in the model would require modelling each school level separately. The use of three school levels provides the currently missing evidence from all educational trajectories (vocational schools very seldom being included in the comparisons). 

5. The article takes a multidimension approach to measuring social connectedness. Even though there is theoretical support for choosing these three indicators of social connectedness (i.e., loneliness, number of friends, belonging), I would like to invite the authors to show empirically using a principal components analysis/ exploratory factor analysis that these three indicators can actually measure a single concept of social connectedness. This is particularly important since the authors state themselves (page 3, line 44) that the measurement of social connectedness is debated.

-Thank you. These three measures are not used together as a scale or latent variable in the current study. Each model is separate for each outcome and allows the interpretation to differ. To our knowledge, these three measures have not been used as a scale in the previous studies either. The question of whether these concepts are correlated is however valid. We have added a short description of the correlations between the items in the results section, pp. 9-10. The magnitude of the correlations between loneliness, number of friends and belonging ranged from small (.33) to medium (-.44) suggesting that these dimensions are related, but different constructs. 

6. The authors mention the use of robust standard errors but I am uncertain whether the regression models account for the clustering of the students within classrooms and schools as well. What was the intra-class correlation coefficient for each sampling level?

-Thank you for this question. The current models do not include information on clustering by classroom or school. We did not have access to these data at the point of analysing the data and reporting. We will study the school-specific factors in the future. This requires data linkage with other registries. In this work we will also investigate the multilevel structure of the dataset. We mention this in the discussion section (p. 24) 

7. Following up to my comment above, I would also appreciate a clearer description of the sampling design (e.g., detailed explanation of stratification, clustering, randomisation, if applicable).

-Thank you for the comment. The SHP study is a national bi-annual survey in Finnish schools. It covers the whole age cohort in the participating age groups. The data is not a sample but a census study. The data used in this study was large, covering about three-quarters of the whole cohort and including answers from almost every Finnish school. We have added more information of the study design in the sample and data collection section and two more references to the description of the datasets (p. 7). 

8. It would be great if the authors could provide the Little’s MCAR test to and examine the missing values’ patterns as well. Listwise deletion is unlikely to affect the estimates in such a big sample, but for the sake of thoroughness, I would advise considering this.

-Thank you. We have added MCAR test in the section for reporting missingness (p. 7). We also added a supplementary file on the detailed proportions of missingness in each combination of the background factor and outcome (Online resource 3). 

9. Ordered logistic regression assumes proportional odds/ parallel regression. Also, there are other regression assumptions, such as residual normality or the absence of multicollinearity. Meeting assumptions has not been mentioned inside the manuscript, but it is important to verify that the model is robust to these before proceeding with the key results.

-Thank you. We have added the diagnostics of the parallel lines for ordered logistic regression, residuals and collinearity as supplementary material (Online resource 3) and mention this in text (p. 9). Note that not all assumptions of linear regression can be applied to ordered logistic and generalised linear models with different distributions. There are also limitations on using the postestimation methods with large sample sizes. We display the postestimations for those that are relevant for each outcome. Using robust standard errors (VCE) includes the robustness to heteroskedasticity in the model.

10. There is a good discussion of the findings in light of previous evidence. Meanwhile, I do certainly feel that the discussion section begins rather abruptly (lines 375-379). The readers would appreciate a smoother recap of the aims of the study, the context a bit, the strengths and the key contributions.

-Thank you. We have added a recap of the aims at the start of the discussion section (p. 18). 

11. What is missing from the discussion section is a brief paragraph/subsection on the policy and practice implications of the findings.

-Thank you for pointing this out. We discuss more policy and practice implications in the conclusions section, pp. 25-26. 

At this point I would like to invite the authors to proofread the paper once more for clarity and conciseness. For example, “Social connectedness is nevertheless not a uniform concept”. I am not certain that the term ‘uniform concept’ is clearly explaining what the authors wish here. “during school years” during the school years (page 3). This sentence is hard to follow because the syntax is confused: “For instance, Högberg et al. [18] reported that although school belongingness declined most in foreign-born students, students from disadvantaged background and low achieving students in Sweden between 200-2018, initially in 2000 there were little differences in school belongingness by these background characteristics.” Another typo: “Because of the selected measured used”

I do not have the time to do a proofreading throughout the manuscript but please take some time to review the language as well.

-Thank you for careful reading. We have proofread the revised manuscript and corrected any typos, including the ones pointed by the reviewer.

---

## [Decision Letter · Decision Letter 1]

10 Oct 2024

The cohort trends of social connectedness in secondary school students in Finland between 2017 and 2021

PONE-D-24-04572R1

Dear Dr. Read,

We’re pleased to inform you that your manuscript has been judged scientifically suitable for publication and will be formally accepted for publication once it meets all outstanding technical requirements.

Kind regards,

Asami Shinohara

Academic Editor

PLOS ONE

Additional Editor Comments (optional):

Reviewers' comments:

Reviewer's Responses to Questions

**Comments to the Author**

1. If the authors have adequately addressed your comments raised in a previous round of review and you feel that this manuscript is now acceptable for publication, you may indicate that here to bypass the “Comments to the Author” section, enter your conflict of interest statement in the “Confidential to Editor” section, and submit your "Accept" recommendation.

Reviewer #1: All comments have been addressed

Reviewer #2: All comments have been addressed

2. Is the manuscript technically sound, and do the data support the conclusions?

Reviewer #1: Yes

Reviewer #2: Yes

3. Has the statistical analysis been performed appropriately and rigorously? 

Reviewer #1: Yes

Reviewer #2: Yes

4. Have the authors made all data underlying the findings in their manuscript fully available?

Reviewer #1: No

Reviewer #2: Yes

5. Is the manuscript presented in an intelligible fashion and written in standard English?

Reviewer #1: Yes

Reviewer #2: Yes

6. Review Comments to the Author

Reviewer #1: Thank you for the opportunity to review a revised version of this manuscript. I am satisfied that all comments and suggestions have been addressed by the authors, and recommend this paper be accepted for publication.

Reviewer #2: The authors have done an excellent job in addressig my initial comments. The manuscript can now be published in its current form, according to my opinion. Best wishes, the reviewer

7. PLOS authors have the option to publish the peer review history of their article (what does this mean?). If published, this will include your full peer review and any attached files.

Reviewer #1: No

Reviewer #2: **Yes: **Dr Ioannis G. Katsantonis

---

## [Editor Report · Acceptance letter]

16 Oct 2024

PONE-D-24-04572R1 

PLOS ONE

Dear Dr. Read, 

I'm pleased to inform you that your manuscript has been deemed suitable for publication in PLOS ONE. Congratulations! Your manuscript is now being handed over to our production team.

Kind regards, 

on behalf of

Dr. Asami Shinohara 

Academic Editor

PLOS ONE